



# Drivers of Droplet Formation in East Mediterranean Orographic Clouds

Romanos Foskinis[1,2,3,4], Ghislain Motos[3], Maria I. Gini[4], Olga Zografou[4], Kunfeng Gao[3], Stergios Vratolis[4], Konstantinos Granakis[4,5], Ville Vakkari[6,7], Kalliopi Violaki[3], Andreas Aktypis[2], Christos Kaltsonoudis[2], Zongbo Shi[8,9], Mika Komppula[10], Spyros N. Pandis[2,11], Konstantinos Eleftheriadis[4], Alexandros Papayannis[1,3], and Athanasios Nenes[2,3*]

[1]Laser Remote Sensing Unit (LRSU), Physics Department, National Technical University of Athens, GR-15780 Zografou, Greece.

[2]Center for Studies of Air Quality and Climate Change, Institute of Chemical Engineering Sciences, Foundation for Research and Technology Hellas, Patras, GR-26504, Greece.

[3]Laboratory of Atmospheric Processes and their Impacts, School of Architecture, Civil & Environmental Engineering, École Polytechnique Fédérale de Lausanne, Lausanne, CH-1015, Switzerland.

[4]ENvironmental Radioactivity & Aerosol Technology for atmospheric & Climate ImpacT Lab, INRASTES, NCSR Demokritos, 15310 Ag. Paraskevi, Attica, Greece.

[5]Climate and Climatic Change Group, Section of Environmental Physics and Meteorology, Department of Physics, National and Kapodistrian University of Athens, Athens, Greece.

[6]Finnish Meteorological Institute, Helsinki, FI-00101, Finland.

[7]Atmospheric Chemistry Research Group, Chemical Resource Beneficiation, North-West University, Potchefstroom, South Africa.

[8]School of Geography, Earth and Environmental Sciences, University of Birmingham, UK.

[9]Key Laboratory of Environmental Optics and Technology, Anhui Institutes of Optics and Fine Mechanics, Chinese Academy of Sciences, Hefei 230031, China

[10]Finnish Meteorological Institute, Kuopio, FI-70211, Finland.

[11]Department of Chemical Engineering, University of Patras, Patras, Greece.

*Correspondence to*: Athanasios Nenes (athanasios.nenes@epfl.ch) and Alexandros Papayiannis (apdlidar@mail.ntua.gr)

## Abstract.

The purpose of this study is to understand the drivers of cloud droplet formation in orographic clouds. We used a combination of modeling, *in-situ* and remote sensing measurements at the high-altitude Helmos Hellenic Atmospheric Aerosol and Climate Change station ((HAC)²), which is located at the top of Mt. Helmos (1314 metres above sea level), Greece during the Cloud-AerosoL InteractionS in the Helmos background TropOsphere (CALISTHO) campaign in Fall 2021 (https://calishto.panacea-ri.gr/) to examine the origins of the aerosols (i.e., local aerosol from the Planetary Boundary Layer (PBL), or long-range transported aerosol from the Free Tropospheric Layer (FTL) contributing to the Cloud Condensation Nuclei (CCN), their characteristics (hygroscopicity, size distribution and mixing state), as well as the vertical velocities distributions and resulting supersaturations.

We found that the characteristics of the PBL aerosol were considerably different from FTL aerosol and use the aerosol particle number and equivalent mass concentration of the black carbon (*eBC*) in order to determine when the (HAC)² was within the FTL or PBL based on timeseries of the height of the PBL. During the (HAC)² cloud events we sample a mixture of interstitial aerosol and droplet residues, which we characterize using a new approach that utilizes the *in-situ* droplet measurements to determine time periods where the aerosol sample is purely interstitial. From the dataset we determine the properties (size distribution and hygroscopicity) of the pre-cloud, activated and interstitial aerosol. The hygroscopicity of activated aerosol is found to be higher than that of the interstitial or pre-cloud aerosol. A series of closure studies with the droplet parameterization shows that cloud droplet concentration ($N_d$) and



supersaturation can be predicted to within 25% of observations when the aerosol size distributions correspond to pre-cloud conditions. Analysis of the characteristic supersaturation of each aerosol population indicates that droplet formation in clouds is aerosol-limited when formed in FTL airmasses – hence droplet formation is driven by aerosol variations, while clouds formed in the PBL tend to be velocity limited and droplet variations are driven by fluctuations in vertical velocity. Given that the cloud dynamics do not vary significantly between airmasses, the variation in aerosol concentration and type is mostly responsible for these shifts in cloud microphysical state and sensitivity to aerosol. With these insights, remote sensing of cloud droplets in such clouds can be used to infer either CCN spectra (when in the FTL) or vertical velocity (when in the PBL). In conclusion, we show that a coordinated measurement of aerosol and cloud properties, together with the novel analysis approaches presented here allow for the determination of the drivers of droplet formation in orographic clouds and their sensitivity to aerosol and vertical velocity variations.

## 1.      Introduction

Aerosol-cloud interactions are holding the largest source of uncertainty in predictions of anthropogenic climate change (IPCC, 2023). A large fraction of this uncertainty arises from impacts of aerosols on cloud droplet formation in liquid and mixed phase clouds (Boucher et al., 2013; Lohmann, 2017). High aerosol levels generally lead to increased cloud droplet number and cloud albedo (Twomey, 1974, 1991), but the exact relationship depends on many factors, including cloud dynamics (cloud scale vertical velocity distributions), aerosol size distribution and hygroscopicity, while the description of these dependencies in a realistic way in models poses a challenge that the development of large observation datasets can help resolve.

Not all clouds are equally sensitive to changes in the cloud condensation nuclei (CCN), i.e., the subset of aerosol that activates into cloud droplets. For clouds to be sensitive to aerosol variations, there needs to be sufficient supersaturation during the stages of cloud droplet formation (Nenes et al., 2001) so that droplet formation can take place. At relatively low concentration of aerosols, water vapor availability (i.e, supersaturation) is large, so that variations in pre-cloud aerosol readily translates to droplet variations. These conditions correspond to "aerosol-limited" clouds, and the $N_d$ is very sensitive to aerosol changes. When CCN concentrations become large, the competition for water vapor required to activate them to cloud droplets becomes so significant, that supersaturation is low and $N_d$ becomes insensitive to aerosol load changes. Under such conditions, clouds are said to be "velocity-limited" (Reutter et al., 2009; Georgakaki et al., 2021), because vertical velocity is the driver of expansion cooling that generates supersaturation. In cases of extreme competition for water vapor, droplet number tends to reach a "limiting" value that is solely a function of vertical velocity (e.g., Georgakaki et al., 2021).

The differences between the aerosol that is involved in cloud droplet formation (i.e., the CCN), and those that do not, called "interstitial", are important to understand. Studies focusing on both the activated and the interstitial particles have been carried out on airborne platforms (e.g., Ditas et al., 2012; Kleinman et al., 2012), and in high-altitude stations (Collaud Coen et al., 2018) that can reside in clouds formed on mountain tops, such as Puy de Dome (Venzac et al., 2009; Asmi et al., 2012), Jungfraujoch (Hammer et al., 2014; Bukowiecki et al., 2016), Storm Peak (Obrist et al., 2008), Mont Sonnblick (Schauer et al., 2016), Mont de Cimone (Marinoni et al., 2008; Cristofanelli et al., 2016), and Zeppelin Station (Tunved et al., 2013).

The established way to separate interstitial aerosols from cloud droplets is to use a "twin inlet system", one of which is used for sampling the interstitial aerosols ("interstitial inlet") and the other for sampling the interstitial and the evaporated cloud droplets ("total" or "whole air inlet"). This sampling strategy is based on the fact that droplet sizes differ substantially from the interstitial aerosols, so an appropriate selection of inlet cut-off size for the interstitial inlet allows for the separate collection of interstitial



aerosols. The challenge is therefore to correctly select the cut-off diameter/size to avoid mixing droplets with interstitial aerosols in the interstitial inlet, given that droplet size varies considerably between clouds. Hammer et al. (2014) and Krüger et al. (2014) used an interstitial inlet consisting of a cyclone with 2.5 µm cut-off diameter ($PM_{2.5}$) to remove droplets, while Portin et al. (2014) and Väisänen et al. (2016), used a $PM_1$ (1 µm cut-off diameter) impactor nozzle plate to prevent the cloud droplets from entering the sample line. Other studies, such as that of Mertes et al. (2005), Drewnick et al. (2006) and Asmi et al. (2012) used a 5 µm cut-off diameter inlet system. Given the large variability in droplet and aerosol sizes, the use of a fixed cut-off size in the interstitial inlet may not always sufficiently separate interstitial aerosols from some evaporated cloud droplets. This can be a problem in velocity-limited clouds with low supersaturation, where interstitial aerosol may have comparable size to the activated droplets (Charlson et al., 2001).

It may also be possible to separate interstitial aerosols from evaporated cloud droplets with a single-inlet (total) system if there are concurrent *in-situ* measurements of droplet size that allows the application of a temporal filter on the timeseries (*e.g.*, consider only parts of the timeseries for which droplet do not pass through the inlet). We explore this technique at the high-altitude Helmos Hellenic Atmospheric Aerosol and Climate Change station $(HAC)^2$ at Mt. Helmos, Greece during the CALISTHO campaign, and study the factors driving cloud formation at an orographic site. We specifically examine the origins and sources of aerosols contributing to the CCN (*e.g.*, local aerosol in the PBL, or long-range transported aerosol from the FTL), their characteristics (hygroscopicity, size distribution and mixing state), as well as the vertical velocities distributions and resulting supersaturations. Several closure studies are carried out to test the ability to predict the droplet number and supersaturation.

## 2.    Methodology

### 2.1    Measurement Sites

Mt. Helmos is the second highest mountain in the Peloponnese (Southern Greece), peaking at 2355 m above sea level (asl.), while the $(HAC)^2$ station is located at the mountain top (coordinates 37.984076 °N, 22.196115 °E), generally isolated from local human activities, and surrounded, at lower altitudes, by lush forests and pristine alpine landscapes. A second temporary site during CALISTHO, called "Vathia Lakka" (VL), was located at the lee side of the mountain approximately 1.7 km away (coordinates 37.999473°N, 22.193391°E) and 500m below $(HAC)^2$. *In-situ* measurements are available at both $(HAC)^2$ and VL, the latter being used as a pre-cloud or post-cloud proxy.

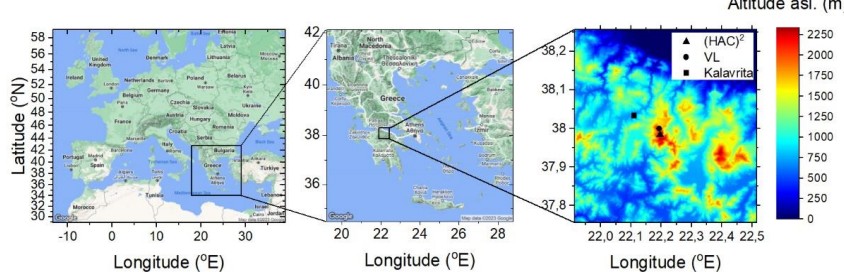

**Figure 1.** The study area (left) and the sub-domain over Greece (middle), and the regional area around HELMOS (right) shown with red color on the alitimeter map given by NASA Digital Elevation Model (NASA JPL, 2020). The symbols present the location of the two sites $(HAC)^2$ and VL, while Kalavrita is the closest village to the site.

Given that the $(HAC)^2$ can be either within the FTL or within the PBL (Foskinis et al., under review), a major parameter controlling the aerosol sampled at any given time is the height of the PBL (PBLH). When



the PBLH exceeds the (HAC)² altitude, the site resides within the PBL, which may be rich in biogenic particles originating from the nearby forest, and anthropogenic emissions originating from the greater region. When the PBLH is below (HAC)², the station is in the FTL and receives airmasses and aerosol from long-range transport: continental aerosols originating from Europe and the Balkans; marine aerosols from the Mediterranean Sea, and dust from Sahara (Papayannis et al., 2005, 2008; Kallos et al., 2007; Kaskaoutis et al., 2012; Soupiona et al., 2018). The (HAC)² frequently resides in the clouds, about ~25% from September to ~65% in October; in November and December the (HAC)² the cloud coverage is 45% (Figure S5). During the whole period the (HAC)² resides about half of the time within the FTL (Foskinis et al., under review).

## 2.2 Dataset and Study Period

The dataset analyzed was collected during the CALISTHO campaign, which was designed to study the cloud microphysical properties using *in-situ* and remote sensing techniques. We focus on October and November 2021, dividing the study period into PBL-influenced and FTL-influenced regimes based on the PBLH timeseries by Foskinis et al. (under review). Furthermore, we divide each regime into "Cloudy" when the cloud Liquid Water Content (LWC) exceeded 0.02 g m⁻³ (Prabhakar et al., 2014; Braun et al., 2018; Dadashazar et al., 2018), and "Cloud Free" otherwise.

## 2.2 Instrumentation

### 2.2.1 Scanning Mobility Particle Sizer (SMPS)

A Scanning Mobility Particle Sizer (SMPS) was used to measure submicron aerosol size distributions at (HAC)², every 5 minutes. The SMPS employs a Vienna-type DMA (electrode length 28 cm) with a condensation particle counter (CPC model 3772, TSI Inc.) to measure particles ranging from 10 to 800 nm. The SMPS operated at a sheath flow rate of 5 L min⁻¹ and an aerosol flow rate of 1 L min⁻¹. Before detection, the ambient aerosol enters the DMA and passes through an 85Kr neutralizer to achieve an equilibrium charge distribution. Both the aerosol sample flow and the sheath air flow were dried below 40% relative humidity using Nafion dryers. The temperature, relative humidity, and pressure inside the instrument are continuously monitored during the sampling process. Additionally, there was a second SMPS (DMA model 3081 and CPC model 3775, TSI Inc.) located at VL and was operated at a sheath flow rate of 3 L min⁻¹ and an aerosol flow rate of 0.3 L min⁻¹, and equipped with a Nafion dryer to dry the sheath air flow below 40% relative humidity. The second SMPS was used to provide the pre-cloud proxy in the cases where (HAC)² was fully covered by clouds.

### 2.2.2 Time-of-Flight-Aerosol Chemical Speciation Monitor (ToF-ACSM)

The ToF-ACSM (Aerodyne Research Inc., Billerica, MA, USA) measures the non-refractory submicron aerosol mass and chemical composition (ammonium, sulfate, nitrate, chloride, and organics) every 10 minutes (Fröhlich et al., 2013). The sampling air enters through a PM₂.₅ virtual impactor, which is followed by a Nafion drier. A 120 μm orifice (for high-altitude measurements) was used for sampling the PM₁ fraction. An aerodynamic particle focusing lens converts the sample into an air beam under high vacuum conditions. The non-refractory material is then flash vaporized on a tungsten plate surface at 600 °C and is subsequently ionized by electron impact ionization at 70 eV. The resulting ions are detected by a TOFWERK time-of-flight mass analyzer. The instrument allows the detection of aerosols of ~40-1000 nm vacuum aerodynamic diameters. A collection efficiency CE of 0.3 (Zografou et al., under review), was chosen based on comparison of the PM₁ mass as the sum of the ACSM and *eBC* concentrations versus the total PM₁ mass by the SMPS (Fröhlich et al., 2015) using densities of 1.8 g cm⁻³ for sulfates and 1.3 g cm⁻³ for organics. The CE accounts for the fraction of the non-refractory particles that bounce off the vaporizer and are not detected. The Relative Ionization Efficiencies for sulfate and ammonium was determined after the calibration of the instrument with ammonium sulfate and ammonium nitrate to be 1.19 and 3.11, respectively.



### 2.2.3    Aethalometer (AE31)

A seven wavelength (370, 470, 520, 590, 660, 880, and 950nm) Aethalometer (AE31 model, Magee Scientific) measures absorption by light-transmission measurements through a filter tape and was used to determine the *eBC* concentration following Hansen et al. (1982) and Petzold et al. (2013). We used the 880 nm channel to constrain the equivalent black carbon concentration. *eBC* is used in this study as a FTL proxy when *eBC*, while when it is high proxy as PBL, since black carbon mostly originates from

anthropogenic activities, and thus its concentration is expected to be much higher in the PBL than in the FTL (Lund et al., 2018; Motos et al., 2020).

### 2.2.4    Particulate Volume Monitor (PVM-100)

A Particulate Volume Monitor (Gerber Scientific Inc, PVM-100) was used to obtain the Liquid Water

Content (LWC), the Particle Surface Area (PSA) and the Effective Droplet Radius ($R_{eff}$) and Diameter ($D_{eff}$) of clouds by measuring the forward scatter of droplets encountered by a diode laser beam along a 40-cm path (Gerber, 1984) in an open path geometry. The signals are then converted to droplet size and number concentration as follows: $N_d = 1.07 \frac{LWC}{\rho_w R_{eff}{}^3}$ (Rezacova et al., 2007).

### 2.2.5    Cloud Condensation Nuclei Counter (DMT CCN-100)

CCN concentrations as a function of supersaturation ("CCN spectra") were measured with a Droplet Measurement Technologies CCN-100 counter, which is based on the Continuous Flow Streamwise Thermal Gradient Chamber design of Roberts and Nenes (2005). The instrument generates supersaturation through the principle of relative diffusion of heat and water vapor. An aerosol sample, surrounded by filtered sheath air, flows through a cylindrical, metallic tube in the axial direction with

wetted inner walls and a positive streamwise temperature gradient. The relative diffusion of water and heat from the tube walls towards the centerline generates a supersaturation (*s*) that peaks at the centerline. The value of this maximum supersaturation depends on the flow rate, streamwise temperature gradient and pressure (Roberts and Nenes, 2005; Lance et al., 2006). Part of the aerosol sample, which is mostly located around the centerline, becomes activated and grows to large enough sizes (0.75 – 10 µm diameter)

to be detected at the exit of the flow tube by an Optical Particle Counter (OPC). The CCN concentration then at the centerline *s* is equal to the number concentration of droplets measured in the OPC. By changing the streamwise temperature gradient every 6 minutes (and ignoring data collected during the instrument transients), we cycle through 0.1% up to 1% with a supersaturation step of 0.1% to obtain a CCN spectrum every hour.

### 2.2.6    Wind Doppler lidar

The vertical velocity of the air was derived by using a wind Doppler lidar (StreamLine XR, HALO Photonics) operating, in stare mode (Pearson et al., 2009). It was installed at the VL site, in order to obtain the updraft currents towards the (HAC)[2] and the surrounding area. Excluding precipitation, the HALO provides vertical velocity (*w*) at 30 m range resolution. Range of the Halo StreamLine XR lidar is 12 km,

but during the campaign the maximum range of useful signal varies from 2 to 3 km depending on the atmospheric aerosol load. Vertical stare was configured at 5 s integration time, alternating between co- and cross-polar receiver. In addition to the vertical stare, Velocity Azimuth Display (VAD) scans were included but are not utilized here. The HALO is a pulsed Doppler lidar and operates at 1.5 µm wavelength (Pearson et al., 2009). The backscattered frequency of each pulse shifts due to the "Doppler effect", which

depends on the relative motion of the scatterer and HALO (Newsom and Krishnamurthy, 2020). The backscattered fraction of the initial pulse is collected back from HALO and analyzed as a time- and frequency-resolved signal. The time delay between each outgoing and backscattered pulse indicates the distance of the scatterers, while the Doppler shift reveals their radial velocity of the scatterer, which corresponds to the aerosol velocity at the given height.



Following Barlow et al. (2011), Newsom and Krishnamurthy (2020) we excluded the data with Signal-to-Noise Ratio (SNR) lower than -20 dB, which limits instrumental uncertainty in $w$ to 0.1 m s⁻¹ at maximum (Pearson et al., 2009). Then, we generated segments containing datasets with 30 minutes moving window (Schween et al., 2014) of the noise-filtered dataset for every 5 min (Lenschow et al., 2012), and we calculated the standard deviation of $w$ ($\sigma_w$) for every height. Considering that a convective plume within

the PBL has on average an ascent speed of 1 m s⁻¹ and that the typical mixing layer height in our site is about 1 km, then the average interval is about twice the lifetime of the plume. This is typical for the derivation of turbulent fluxes from eddy covariance stations (Schween et al., 2014). The 30 min window is comparable to the average mixing time in the boundary layer.

### 2.3 Modelling

#### 2.3.1 Aerosol hygroscopicity and critical supersaturation

Two approaches are used to constrain the aerosol hygroscopicity. First, we determine the bulk hygroscopicity parameter ($\kappa$) of the submicron aerosol (Petters and Kreidenweis, 2007) using the measured chemical composition and the approach outlined in Padró et al. (2010). This involves applying the ISORROPIA II thermodynamic equilibrium model (Fountoukis and Nenes, 2007) using as inputs the

observed inorganic components measured by ToF-ACSM, to calculate the composition (e.g., $NH_4HSO_4$, $NH_4Cl$, $NH_4NO_3$, $(NH_4)_2SO_4$, $H_2SO_4$) of the inorganic aerosol fraction. The Zdanovskii, Stokes and Robinson (ZSR) mixing rule was then applied on the volume fraction of the inorganic salts, including the volume fraction of $eBC$ and hygroscopicity parameters from Table 2 of Padró et al. (2010), while the hygroscopicity of value of $eBC$ was considered equal to 0.2 based on Ding et al. (2021). Second, we

determined the characteristic hygroscopicity parameter ($\kappa^*$) which was explicitly obtained from the combination of the CCN-100 and SMPS data based on the CCN spectrum cycles. For each supersaturation cycle we calculated the characteristic critical supersaturation ($s^*$) (Cerully et al., 2011), which is defined as the supersaturation at which half of the CCN population is activated to droplets and is determined using the procedure of Cerully et al. (2011), and the characteristic size ($D_{cr}^*$), which is defined so the

corresponding SMPS distribution integrated from the largest resolved size (800 nm) until the $D_{cr}^*$ to give an aerosol number equal to the observed CCN concentration. Then, the $\kappa^*$ of each CCN cycle was determined from $\kappa$-Kölher theory (Petters and Kreidenweis, 2007) as $\kappa^* = \left( \frac{4 A^3}{27 D_{cr}^{*3} s^{*2}} \right)$, where $A = \frac{4 M_w \sigma}{R T \rho_w}$ is the Kelvin parameter, $M_w$ (kg mol⁻¹) is the molar mass of water, $\sigma$ (J m⁻²) is the surface tension of the activated droplets (here assumed to be equal to pure water), $R$ (J mol⁻¹ K⁻¹) is the universal gas constant, $T$

(K) is the ambient temperature, $\rho_w$ is the density of liquid water, and $D_{cr}^*$ is the characteristic dry size of the particle that activates at supersaturation $s^*$.

Both approaches give similar results (not shown here) but occasionally diverge. For instance, the bulk hygroscopicity assumes that particles are internally-mixed, which for periods of sampling in the FTL is an excellent assumption (e.g., Bougiatioti et al., 2016) while for PBL-dominated periods it is also a

reasonable assumption due to the remote location of the station while the characteristic hygroscopicity is strictly corresponded to the average hygroscopicity for particles of size $D_{cr}^*$, and was used to characterize the hygroscopicity of the resolved CCN spectrum. During periods where particles at the station will originate from both PBL and FTL, it is expected that they have an intermediate mixing state that will introduce some uncertainty in subsequent calculations.

#### 2.3.2 Droplet Activation Parameterization

We use a physically based aerosol activation parameterization developed by Nenes and Seinfeld (2003) and further expanded by Fountoukis and Nenes (2005), Barahona et al., (2010) and Morales Betancourt and Nenes (2014) to calculate the droplet number formed in clouds using the wind vertical velocity and the aerosol characteristics. The droplet activation parameterization solves the equations of motion of an

ascending air parcel which contains aerosols and water vapor and calculates the point where the supersaturation maximizes ($s_{max}$) as well as the corresponding droplet number, $N_d$.



The parameterization inputs include pressure, temperature, aerosol size distribution measured by SMPS, the bulk hygroscopicity parameter $\kappa$, and the updraft velocity obtained by HALO. Given that the latter varies considerably over time and within each cloud event, we consider a probabilistic approach by first computing the hourly Probability Density Function (PDF) of vertical velocity. We then apply the parameterization to calculate the PDF-integrated $N_d$, which is assumed to describe the average droplet number in clouds that form in the vicinity of the sampling site. This PDF-averaging approach has been shown to successfully reproduce cloud-scale values of $N_d$ in numerous field studies in case of cumulus and stratocumulus clouds (Conant et al., 2004; Meskhidze et al., 2005; Fountoukis and Nenes, 2007; Kacarab et al., 2020; Georgakaki et al., 2021; Foskinis et al., 2022).

The PDF-integrated $N_d$ is computed using the characteristic velocity ($w^*$) approach of Morales and Nenes (2010), in which the parameterization is applied once using $w^*$ in its input but provides directly the PDF-averaged droplet number. $w^*$ is given by (Foskinis et al., 2022):

$$w^* = \varepsilon \, \widetilde{\lambda} \, \sigma_w$$

where $\sigma_w$ is the standard deviation of the vertical velocity PDF (assumed to be a Gaussian with zero mean), $\varepsilon$ is the entrainment parameter and $\widetilde{\lambda}$ the characteristic nondimensional velocity. The entrainment parameter ($\varepsilon$) accounts for lateral diabatic mixing of entrained air in the updraft zones – which effectively reduces $s_{max}$ hence $w^*$. $\varepsilon = 1$ corresponds to adiabatic updrafts, but the parameter can be lower in the case of cumulus and convective clouds, affecting the vertical distribution of liquid water and number of droplets (Morales et al., 2011). Based on numerous *in-situ* sampling campaigns of boundary layer clouds, $\varepsilon$ has been was found to be on average 0.68. According to Morales and Nenes (2010), $\widetilde{\lambda}$ is affected by the total aerosol concentration ($N_{Total}$), and it is assigned values equal to 0.70, 0.79, 0.84 and 0.98, when $N_{Total}$ ranges between 0-340 cm$^{-3}$, 340-500 cm$^{-3}$, 500-6400 cm$^{-3}$ and 6400-106000 cm$^{-3}$, respectively.

## 3. Experimental Results

### 3.1 Dataset Overview

Figure 2a shows when (HAC)² is in the FTL and when in the PBL based on the relative position of the PBLH (given by Foskinis et al., under review) against the altitude of the (HAC)² station (shown in black horizontal line). When the station is in the FTL, the PBLH is below (HAC)² line and vice versa when it is in the PBL. The airmass typing is consistent with observed the moisture content, since under cloud-free conditions, the FTL airmasses are markedly dryer (RH=34 ± 26 %) than the PBL airmasses (RH=65 ± 16 %) (Figure 2c). Also, when the (HAC)² is in the FTL, we observed two dominant wind directions, one at 30° N and one at 80° N, where for both $N_{Total}$ approaches the lowest values observed (~45 cm$^{-3}$) (Figure S2d), and $eBC$ levels approach its detection limit (~0.01 µg m$^{-3}$). These wind directions are directly related to the long-range transported airmass; when arriving from the north, it usually originates from E. Europe and the Balkans, and is rich in sulfur (Stavroulas et al., 2021). When the airmass arrives from the E or SE, it often carries dust aerosols. When the (HAC)² is within the PBL, we identified three prevailing wind directions, that correspond to the local transport patterns (Figure S2f) from 90°, 180° and 320° N, where the $N_{Total}$ obtains its maximum values (~3300 cm$^{-3}$) (Figure S2d), and the $eBC$ values increase up to ~0.4 µg m$^{-3}$ when the wind speed exceeds 6 m s$^{-1}$ and becomes maximum when the wind blows from 160 - 220° (Figure S2c).

Additionally, we found a dependence of the PBLH on the wind direction, since when the wind passes over mountain-tops before reaching the site, the PBLH tends to be higher (Figure S2a) and the $\sigma_w$ tends to be lower (Figure S2b). We observed that the increase of the aerosol content (from ~250 cm$^{-3}$ to ~750 cm$^{-3}$) (Figure S2b) leads to an increase of $N_d$ (from ~100 cm$^{-3}$ to ~300 cm$^{-3}$) (Figure S2c), and decrease of the



cloud droplet size (from ~17.5 μm to ~10 μm) (Figure S2d), consistent with the Twomey effect (Twomey,
1977) of aerosols on clouds and cloud albedo (IPCC, 2023).

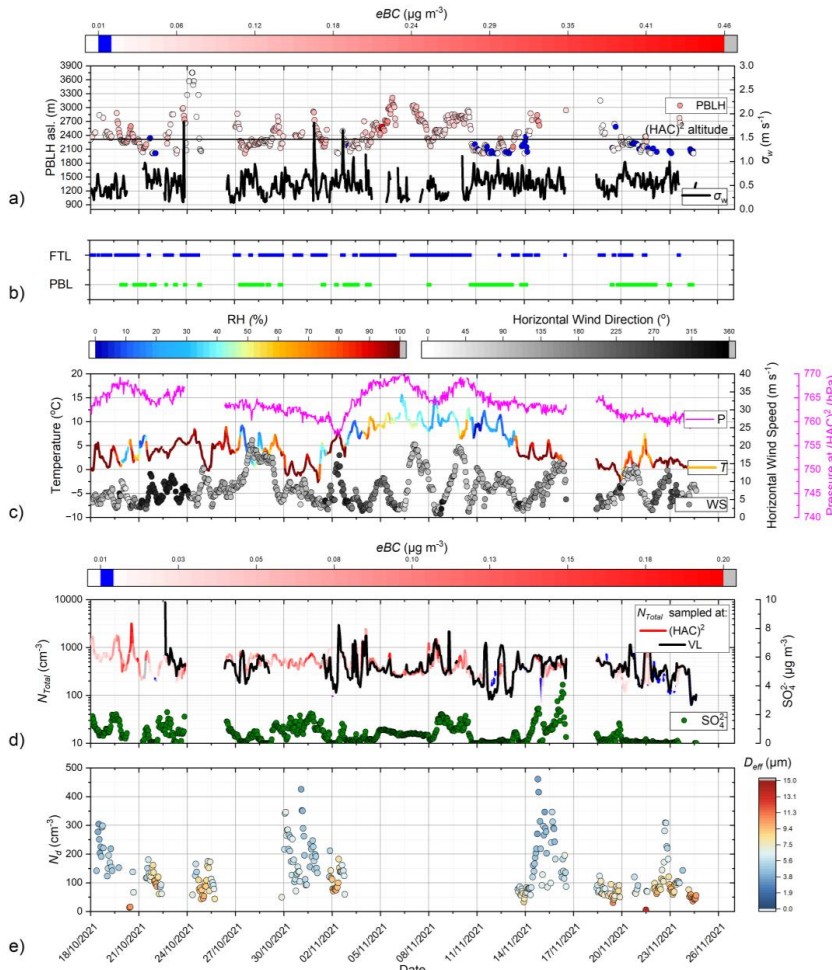

**Figure 2.** Timeseries of characteristic parameters for the CALISHTO campaign. a) PBLH (circles) and filled-color
corresponds to the *eBC*. If PBLH is below the (HAC)² (black horizontal line), the (HAC)² is within the FTL (*eBC* < 0.01
μg m⁻³ – blue color), and vice versa if in the PBL (*eBC* > 0.01 μg m⁻³, red color). On the right-hand side axis, the $\sigma_w$ which
drives the PBLH. b) The moments where (HAC)² is within FTL – blue color, and within PBL – green color, respectively.
c) The ambient air temperature trace is colored by the RH (in the FTL when < 40% and vice versa when inside the PBL).
The horizontal wind speed is given by the circles colored by the horizontal wind direction in grey scale and the ambient
pressure is presented by the magenta line. d) $N_{Total}$ measured at (HAC)² – colored by the *eBC* amount and the $N_{Total}$
measured at VL – black line. $N_{Total}$ measured at (HAC)² comparable to the one of VL implies both sites reside in the
same atmospheric layer – also indicated by a high concentration of *eBC*. $SO_4^{2-}$ concentration is presented by the green
symbols and is also used as a proxy of the (HAC)² being in the PBL. e) $N_d$ colored by the $D_{eff}$.

### 3.2 Separating Interstitial Aerosol from Cloud Droplet Residuals at (HAC)²

Aerosol particles that act as CCN have a dry diameter of order 100 nm diameter and grow at least 10-fold
when they activate into cloud droplets (Figure 3) (Rogers and Yau, 1996; Pierce et al., 2015). Indeed, when



the (HAC)² station was fully covered by the clouds, the droplet effective diameter, $D_{eff}$, varied between 2 and 15 μm (Figure 2). When the clouds are formed with FTL aerosol, the $D_{eff}$ was on average 17.0 ± 2.7 μm, and 10.3 ± 1.9 μm when formed with PBL aerosol. The average size differences between the two types of airmasses can be explained by the different CCN concentrations in them. The FTL has fewer CCN,

hence droplets are generally larger than in PBL airmasses. In both cases, the aerosol inlet (which is a PM₁₀ inlet - 10μm cut-off diameter) would sample inactivated (known as "interstitial") aerosol together with some of the droplets. These droplets subsequently evaporate in the heated inlet and contribute to the size distribution and other aerosol characteristics measured by the online *in-situ* aerosol instrumentation. This means that when the station is in-cloud, the aerosol sampled from the PM₁₀ inlet corresponds to a mixture

of interstitial aerosol and evaporated cloud droplet residuals. However, $D_{eff}$ varies considerably during a cloud event (Figure 2), and often exceeds 10 μm. This implies that considering subsets of the in-cloud timeseries when the $D_{eff}$ is large enough can ensure that during these periods the PM₁₀ inlet samples only interstitial aerosol, as droplets would be too large to pass through the inlet.

We therefore consider the above concept and develop a "virtual filter" technique to define the $D_{eff}$
threshold (measured *in-situ* and continuously by the PVM-100) that ensures that the aerosol sampled by the PM₁₀ inlet does not contain aerosol from evaporated droplets, but only interstitial aerosols. In applying this filter, we ignore periods of the respective measurements during which the *in-situ* $D_{eff}$ of the droplets is less than the threshold. We select the periods during we were sampling at least 30 minutes continuously in cloud-free conditions followed by (or proceeded by) at least 30 minutes of cloudy conditions, to allow

multiple size distribution measurements during the pre-/post- and the in-cloud phases. The distributions under cloud-free conditions are then averaged, to give the "total aerosol distribution". The in-cloud distributions are averaged for periods where the droplet $D_{eff}$ exceeds a predefined threshold (starting from 10μm). The in-cloud distributions are averaged for different values of the $D_{eff}$ threshold, until 16 μm.

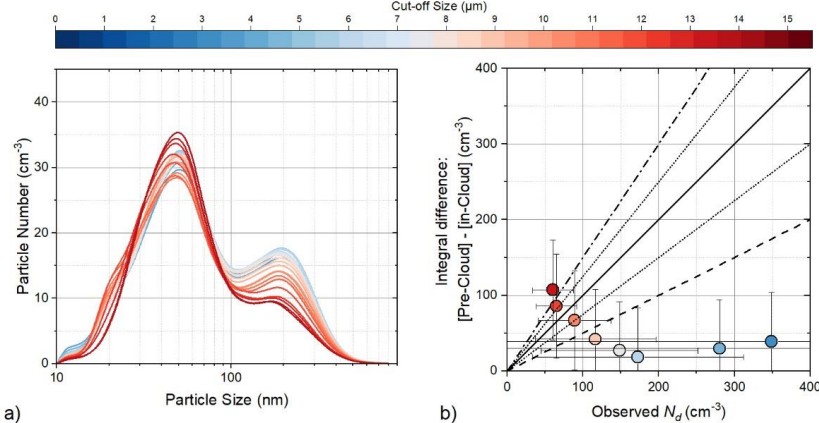

**Figure 3.** a) The results of the sensitivity analysis using differential cut-off size threshold and the droplet size threshold
($D_{eff}$ as measured *in-situ* with the PVM-100) applied is represented by the color-scale. b) The integrated difference between pre-cloud and in-cloud aerosol size distributions from ~70 nm to the largest sizes when compared to the droplet number measured *in-situ* concurrently by the PVM-100. Each symbol corresponds to application of a different $D_{eff}$ threshold (as indicated by the symbol color, using the same scheme as in a), while the errorbars correspond to the standard deviation.

We select as the optimum $D_{eff}$, the minimum value above which the measured aerosol size distribution becomes insensitive to the chosen threshold value. Figure 3 displays an example of this process applied to a segment of data from CALISHTO. We find in this case that a $D_{eff}$ threshold of 13.5 μm is the minimum for which the observed size distribution stopped to be sensitive to the changes in the cut-off size (Figure

4000



3a). Additionally, we compare the difference between the aerosol size distribution pre-cloud and the interstitial aerosol distribution (i.e., with the application of the 13.5 µm threshold) with the observed droplet number, and we found that indeed, the integrated difference between these distributions (from ~70 nm to the largest sizes measured by the SMPS) matches with the droplet number obtained *in-situ* with the PVM-100 to within ±25% (Figure 3b). Thus, we confirm that this threshold is consistent with allowing only interstitial to pass through the inlet. Given that *in-situ* closure studies often involve this degree of uncertainty (e.g., see relevant discussion by Foskinis et al. (under review) and relevant references cited therein) in addition to any other uncertainties that may exist at this particular site (e.g., variations of aerosol entering the cloud, sampling efficiency of the inlet and uncertainties in the droplet number determination with the PVM-100), we conclude that the latter distribution is indeed representative of the interstitial aerosol. Additional support for this is conclusion is provided later by the ability to predict cloud droplet number (section 3.4) as it requires the correct parameters of hygroscopicity, size distribution and vertical velocity.

### 3.3 Differences among the properties of total, activated, and interstitial aerosol

We identified more than 20 periods of cloud-free/cloudy transitions , include during the CALISHTO campaign. We applied the methodology of Section 3.2 to estimate the corresponding cloud-free, interstitial-only, and mixed aerosol (cloud residues and interstitial combined) size distributions. We then determined $s*$ and $\kappa*$ (Section 2.4.1) of the cloud-free ($\kappa_{cf}$), interstitial ($\kappa_i$) and interstitial-droplet residues aerosol mixture (Figure 4a). Assuming that the pre-/post-cloud hygroscopicity is the volume average hygroscopicity of the interstitial and activated aerosol, we estimate the hygroscopicity of the activated aerosol, $\kappa_a$, using the mixing rules of Petters and Kreidenweis (2007) as $\kappa_a = \frac{\kappa_{cf} V_{cf} - \kappa_i V_i}{V_{cf} - V_i}$, where $V_{cf}$ is the total volume of pre-/post-cloud aerosols and $V_i$ is the total volume of interstitial aerosols, respectively. The estimation of $\kappa_a$ assumes that all populations are internally mixed and the activated cloud aerosol plus the interstitial aerosol is equal to the pre-/post-cloud aerosol volume. This is a reasonable assumption given that Brownian losses affect the smallest particle sizes which have a minor contribution to the aerosol volume.

Figure 4 presents the results of our analysis. Figure 4a shows the characteristic supersaturation, $s*$, for each aerosol population. Typically, $s*$ is higher for interstitial aerosol and lower for the mixed. This is consistent with the expectation that particles that activates to form droplets tends to be more hygroscopic than the interstitial aerosol (e.g., Cerully et al., 2011). Indeed, during periods where cloud formation is influenced by FTL airmasses, the average $\kappa*$ was 0.34 ± 0.09 % for pre-/post-cloud, 0.31 ± 0.15% for interstitial, and 0.45 ± 0.20% for activated aerosol. During periods that clouds were forming on PBL aerosol, the average $\kappa*$ was 0.43 ± 0.12% for pre-cloud, 0.29 ± 0.19% for interstitial and 0.44 ± 0.18% for activated aerosols (Figure 4b).

The average $s*$ (Figure 4c), during the periods where cloud formation is influenced by FTL airmasses, was found equal to 0.56 ± 0.21% and 0.59 ± 0.22%, while during PBL influenced periods, was found 0.27 ± 0.18% and 0.28 ± 0.16% for the pre-/post-cloud and interstitial aerosols, respectively. These results showed little sensitivity to airmasses origins, i.e., FTL or PBL. Clearly, the interstitial aerosol is less hygroscopic on average, and the activated aerosol can be up to twice as hygroscopic. This is important for understanding the role of cloud processing on aging of particles and transferring hygroscopic material to evaporated cloud residuals.

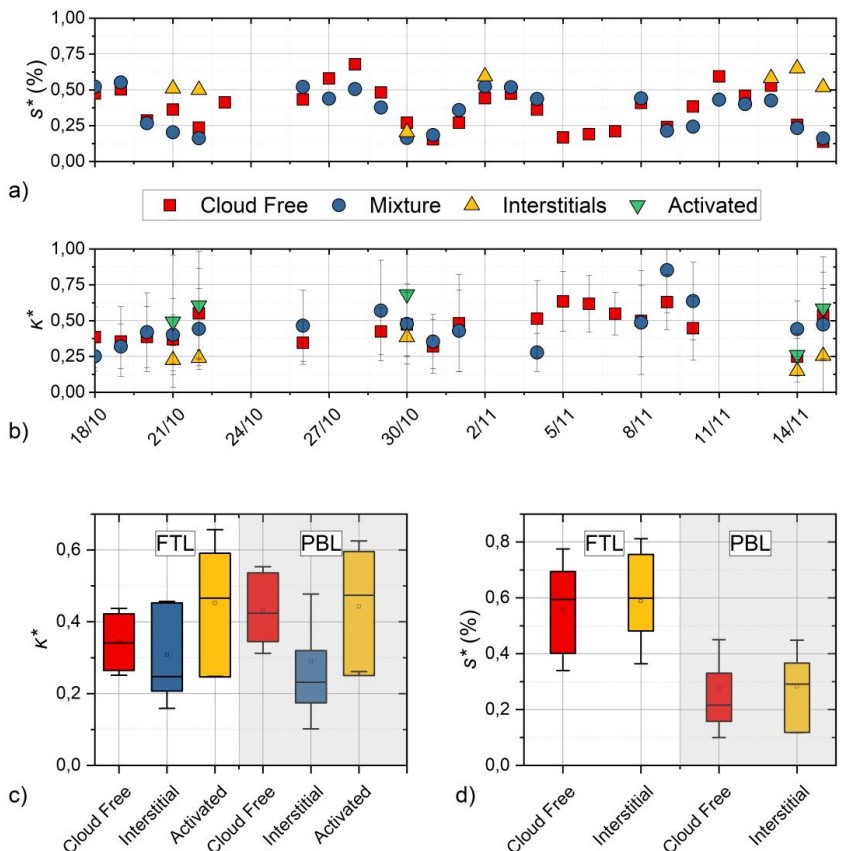

**Figure 4.** Daily averaged a) $s^*$ and b) $\kappa^*$ of the aerosol in cloud free regimes, of the mixture of interstitial aerosols and some droplets residues, and of the activated aerosols; The c) and d) the distributions of $\kappa^*$ and $s^*$ when the airmass originating from the FTL or the PBL, respectively.

Analysis of $s^*$ of the activated aerosol can provide important insights about cloud formation when FTL or PBL airmasses are involved. The critical supersaturation of most of the activated cloud droplet residuals should be close to the maximum supersaturation in the cloud. Indeed, when cloud $s_{max}$ is high, droplet formation is aerosol limited and vice versa when droplet formation is velocity limited. According to Georgakaki et al., (2021) and Motos et al., (2023), clouds are velocity limited when the $s_{max}$ is ~0.15% or

lower, and aerosol-limited otherwise. Indeed, using $s^*$ of the activated aerosol population as a proxy of $s_{max}$ (which is supported by the analysis of Section 3.4) we see that clouds formed from FTL airmasses have $s^* > 0.5\%$, hence the corresponding clouds are highly aerosol-sensitive. In contrast, clouds formed in PBL airmasses have a much lower $s^*$, reaching even 0.15% (Figure 4d) hence their formation tends to be velocity-sensitive. Given that the $\sigma_w$ does not change significantly when clouds form upon FTL or PBL

airmasses ($\sigma_w = 0.58 \pm 0.25$ m s$^{-1}$), and given that the $N_{Total}$ in PBL airmasses was roughly three times higher than the $N_{Total}$ in FTL airmasses (approximately 750cm$^{-3}$ and 250cm$^{-3}$, respectively) (Figure S3a), much of this distinction between aerosol- and velocity-limited conditions is driven by variations in aerosol, rather than variations in cloud dynamics (i.e., $\sigma_w$).





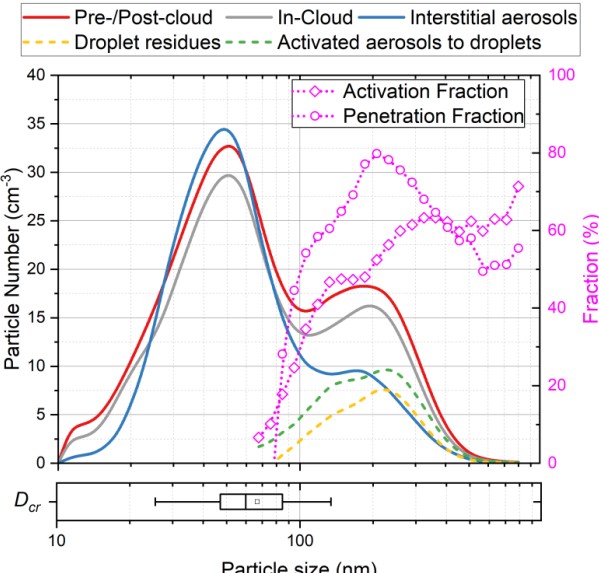

**Figure 5.** Averaged particle size distributions for pre-/post-cloud moments, in-cloud, and interstitial (using the $D_{eff}$ threshold of 13.5 μm). Shown also are two estimations of the activated aerosol distribution, the droplet residues or "dried droplets" distribution (yellow dashed line), estimated from the difference between the measured in-cloud and the interstitial aerosols, and the "activated droplets" distribution (green line) estimated by the difference between the pre-cloud and the interstitial aerosol distribution, respectively. The latter gives an estimate of the aerosol that gives droplets that are too large to be sampled at any size threshold by the inlet, while the former provides the activated aerosol from droplets that are sampled by the inlet when in-cloud. The activation and the penetration fraction were estimated similarly by counting the ratio between the "activated droplets" and "pre-/post-cloud aerosols", and, "dried droplets" to "pre-/post-cloud aerosols", respectively.

Figure 5 shows that is measured on average during the cloud sampling through PM$_{10}$ in respect of the aerosols and the droplet residues. The pre-/post-cloud and the in-cloud datasets here are the same that were used in Section 3.2 (the periods during were we sampling at least 30 minutes continuously in cloud-free conditions followed by (or proceeded by) at least 30 minutes of cloudy conditions) and the interstitial aerosols dataset derived after removing from the in-cloud dataset the data were the $D_{eff}$ was exceeding the threshold of 13.5 μm. Thus, we calculated the average size distributions of the pre-/post- and the in-cloud phases as well as the interstitial aerosols. The average size distribution of aerosols that activated to droplets is then the difference between the average distribution of the pre-/post cloud aerosols and the interstitial aerosol, while the average size distribution of the droplet residuals was derived by the difference of the in-cloud and interstitial aerosols . At last, we calculated the size resolved "activation fraction" as the ratio of the interstitial aerosols to the pre-/post cloud aerosols (from 70nm and above, given that smaller particles are not expected to activate), and the size-resolved "penetration fraction" as the ratio of droplet residues to the in-cloud aerosols. Hence, we found that when we sampling within the clouds through a PM$_{10}$ inlet, the penetration fraction on average can reach up to 80%. This, means that comparison of pre-/post- and in-cloud aerosol distributions may provide qualitatively consistent microphysical insights that are however subject to uncertainty of around 40%. In the end, we calculated the size distribution of the aerosols that activated droplets by the difference between the in-cloud and the interstitial averaged size distributions, and the size-resolved activation fraction as the ratio of the activated aerosols to droplets against the pre-/post- aerosols. We found that the activation fraction is roughly 60% for most of the activated aerosol sizes (Figure 5), while about 50% of them constitute the droplet residues that penetrated the PM$_{10}$ inlet.





### 3.4 Closure Study of $N_d$ and $s^*$

We applied the droplet activation parameterization of Morales Betancourt and Nenes (2014), using the size distributions measured at the $(HAC)^2$ and the VL, and the $\sigma_w$ and bulk hygroscopicity parameter $\kappa$ measured at $(HAC)^2$, to predict the $N_d$ and $s^*$ of the clouds formed at $(HAC)^2$. It is important to mention here that the droplet activation parameterization of Morales Betancourt and Nenes (2014) is designed to calculate the $s_{max}$ and the $N_d$ when it is initialized by the ambient aerosols, and that's why we used the size

distributions that was measured at VL. On the other hand, when the use the in-cloud aerosol size distributions measured at $(HAC)^2$, given that this a combination both of interstitial aerosols and droplet residues, these distributions we have already shown in Section 3.3 that differ each other, which result in underestimations on the $N_d$. Here we examine under which conditions the use of the in-cloud aerosol can give reliable results compared against to the *in-situ* observations of $N_d$ and $s^*$, and to evaluate the internal

consistency of the dataset and analysis carried out in the previous section, as well as to evaluate the ability of the parameterization to predict microphysical quantities for clouds influenced by the types of airmasses (FTL, PBL) considered.

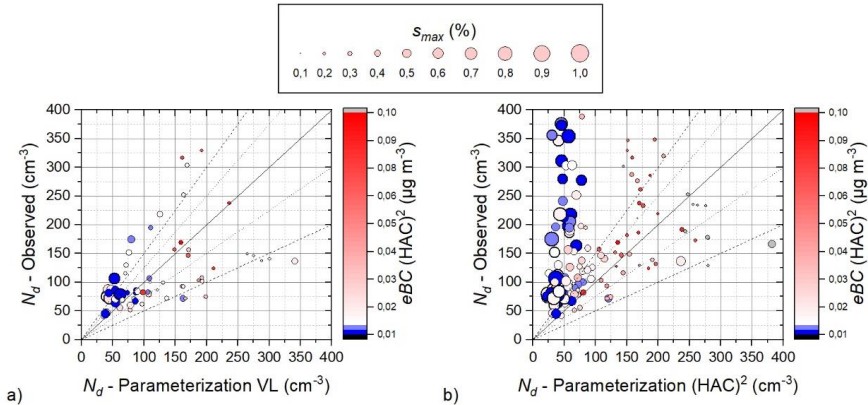

**Figure 6.** a) $N_d$ from PVM-100 observations at the $(HAC)^2$ (vertical axis) against parameterization predictions

(horizontal axis) using observed aerosol distributions from the VL and b) $(HAC)^2$, respectively. The symbol color corresponds to the *eBC* amount, and symbol size corresponds to the parameterization-predicted $s_{max}$. The dashed lines indicate regions of ±25 and ±50% deviation from the 1:1 line.

We found that we can obtain $N_d$ closure to within 25%, when using the aerosol distributions from VL (Figure 6a) - even when the *eBC* levels are low. This may imply that VL may at times also reside in the

470 FTL or at the catabatic region of the cloud during these specific periods, and hence its aerosol distributions may be representative of the total aerosol, including that which activated into cloud droplets. On the other hand, by using the aerosol from $(HAC)^2$ we obtained a reasonable closure only where *eBC* is high and the $s_{max}$ is low– in other words when $(HAC)^2$ cloud droplets were formed upon aerosol from the PBL. However, when aerosol at $(HAC)^2$ is influenced by the PBL (*eBC* is high, more than >0.01 µg m$^{-3}$), $N_d$ is

475 obtained within 25% (Figure 6b). That means that, when the in-cloud aerosol distributions from $(HAC)^2$ are used as input to the parameterization and clouds form in FTL airmasses (i.e., *eBC* is very low, less than <0.01 µg m$^{-3}$), the parameterization highly underestimates $N_d$ (Figure 6b), because activated droplets were not sampled by the PM$_{10}$ inlet. Concluding that the usage of the $(HAC)^2$ distributions lead to underprediction of droplet number (50% or more) especially when the measured *eBC* levels are low,

consistent with the view that VL aerosol is less representative of FTL.

Additionally, we found that when we used the aerosol from VL, $s^*$ agree with $s_{max}$ to within ±25% when *eBC* was high (~0.1 µg m$^{-3}$) (Figure 7a). When we used the aerosol from $(HAC)^2$, the $s^*$ match with $s_{max}$ to within ±25% most of the time (Figure 7b); this is because when in cloud, aerosol exposed at lower



supersaturation values than the $s_{max}$ have already been activated to droplets. Thus, the residuals give $s^*$
values close to $s_{max}$.

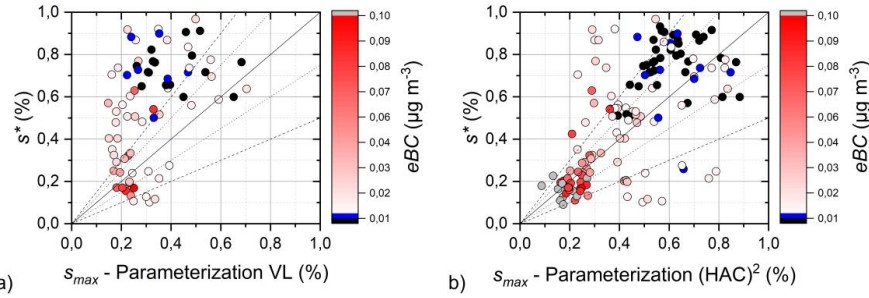

**Figure 7.** a) $s^*$ of the total aerosol distribution (vertical axis) against parameterization predictions (horizontal axis) using observed aerosol distributions from the VL (a; left panel) and b) (HAC)², respectively. The symbol color corresponds to the $eBC$ amount, while the dashed lines indicate regions of ±25 and ±50% deviation from the 1:1 line.

When applying the parameterization to size distributions observed at VL, we expect that the predicted $N_d$ (and $s_{max}$) will be close to observations when the cloud at (HAC)² is dominated by PBL aerosol, and deviate largely when FTL airmasses are at the (HAC)². Similarly, (HAC)² pre-cloud aerosol distributions should provide good predictions of $N_d$ when the airmass at the site is from the FTL. Use of in-cloud aerosol distributions is expected to result in deviations of the predicted from the observed $N_d$ given that nucleation scavenging will lead to cloud droplets that are not sampled by the aerosol inlet, hence will not be measured. The magnitude of this deviation depends on the size of the droplets sampled, which in turn depends also on the amount of aerosol that is available for activation because cloud droplet sizes are expected to become progressively smaller as CCN concentrations increase.

## 4. Conclusions

We study the drivers of cloud droplet formation in orographic clouds using a combination of modeling, *in-situ* and remote sensing measurements at the (HAC)² station during the CALISTHO campaign in Fall 2021. We study the origins of the aerosols, (e.g., local aerosol from the PBL, or from long-range transport from the FTL) which there can be used understand their characteristics (hygroscopicity, size distribution and mixing state), contribution to CCN, $N_d$ and resulting supersaturations.

We found that the $N_{Total}$ and the $eBC$ within the FTL get the low values (~45 cm$^{-3}$ and ~0.01 μg m$^{-3}$), while within the PBL they both get considerably larger values (~3300 cm$^{-3}$ and ~0.4 μg m$^{-3}$). That means that the PBL has more CCN, which result to more (from ~100 cm$^{-3}$ to ~300 cm$^{-3}$) and smaller droplets (from ~17.5 μm to ~10 μm).

We also study the characteristics for aerosol involved in cloud droplet formation, and those that do not activate (i.e., interstitial aerosols). To accomplish this, we develop a new algorithm applied to the aerosol timeseries measured with the PM$_{10}$ single-inlet system, which can sample interstitial aerosol and droplets with size up to the inlet's cut-off size (thus the droplet residues get dried and mixed with the interstitial aerosols) when in cloud. This separation algorithm involves applying a "virtual filter" to the aerosol timeseries from the PM$_{10}$ inlet based on a droplet size threshold (13.5 μm) derived from *in-situ* observations (PVM-100) that determines when the aerosol sampled by does not contain droplet residuals. Thus, when the in-situ average droplet size exceeds the threshold, droplets would be too large to pass through the inlet so that the aerosol sampled by the PM$_{10}$ inlet is interstitial aerosol. Not considering this filter can considerably bias the results, as up to ~ 80% of aerosol can be dried droplet residuals.



By using this approach, we separated the dataset to pre-/post-cloud and interstitial regimes and we studied the characteristics of the pre-/post-cloud, interstitial and activated aerosol to droplets. We found that when airmasses originated from the FTL, the $\kappa^*$ was on average $0.34 \pm 0.09$ % for pre-/post-cloud, $0.31 \pm 0.15$% for interstitial, and $0.45 \pm 0.20$% for activated aerosol When the airmasses originated from the PBL, the $\kappa^*$ was on average $0.43 \pm 0.12$% for pre-cloud, $0.29 \pm 0.19$% for interstitial and $0.44 \pm 0.18$% for activated aerosols, respectively. That means that the interstitial aerosols are the least hygroscopic of aerosol types, and the activated aerosols can be up to twice as hygroscopic than interstitial aerosol. This is important for understanding the role of cloud processing on aging of particles and the accumulation of hygroscopic material on evaporated cloud residuals.

The average $s^*$ during periods where cloud formation is influenced by the FTL, was found equal to $0.56 \pm 0.21$% and $0.59 \pm 0.22$%, while during PBL-influenced periods was $0.27 \pm 0.18$% and $0.28 \pm 0.16$% for the pre-/post-cloud and interstitial aerosols, respectively. These results exhibited little sensitivity to airmasses origins, i.e., FTL or PBL. When a cloud is formed in FTL airmasses, the droplet formation is more sensitive to changes in the aerosol load, while it tends to be more sensitive in changes on vertical velocity when a cloud is formed in the PBL. Given that the cloud dynamics do not vary significantly between airmasses, the variation in aerosol concentration is mostly responsible for these shifts in cloud microphysical state and sensitivity to aerosol.

Finally, a series of closure studies with the droplet parameterization is carried out to determine its ability to predict droplet number, supersaturation and constrain the cloud microphysical state (i.e., whether it is velocity- or aerosol- limited). We show that $N_d$ can be predicted to be within 25% of observations when the aerosol size distributions best approximate the pre-cloud distributions. The high degree of droplet and supersaturation closure ensures that the model-data fusion and novel approaches for determining the aerosol populations (interstitial and activated cloud droplets) are consistent, provide a realistic assessment of cloud state and can be applied in future studies.

In conclusion, we present a coordinated, innovative approach that allows the determination of the drivers of droplet formation in orographic clouds and their sensitivity to aerosol and vertical velocity variations.

*Author contributions.* AN, AP and KE organized the CALISHTO campaign. RF and AN conceived and led this study. RF led the analysis, wrote the original manuscript together with AN and AP, and prepared all the figures with contributions from all authors. RF analyzed the data and interpreted the results with input from AN, AP, and KE. RF, GM, MG, OZ, SV, KG, CK, and AA conducted experiments and collected the raw data. All authors discussed, reviewed and edited the manuscript.

*Financial support.* This work was supported by: PyroTRACH (ERC-2016-COG) funded from H2020-EU.1.1. (ERC), project ID 726165, the Swiss National Science Foundation project 192292, Atmospheric Acidity Interactions with Dust and its Impacts (AAIDI), "PANhellenic infrastructure for Atmospheric Composition and climatE change" (MIS 5021516) projects. AP and RF acknowledge funding by the Basic Research Program PEVE (NTUA) under contract PEVE0011/2021.

*Acknowledgements.* The Biomedical Research Foundation of the Academy of Athens (BRFAA) is acknowledged for providing the mobile platform to host the NTUA AIAS lidar system.

*Competing interests.* The authors declare that they have no conflict of interest.

*Data availability.* The data presented in this publication will be made available from the Swiss EnviDat platform (https://www.envidat.ch). The DOI link will be activated for public access upon acceptance of publication.



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
