# Peer review of "Drivers of Droplet Formation in East Mediterranean Orographic Clouds"

_EGUsphere, 2024_

## Referee Comment (RC1)

Review of "Drivers of Droplet Formation in East Mediterranean Orographic Clouds" by Foskinis et al.

**Summary:**

This paper investigates properties of aerosol and droplet residuals in orographic clouds measured with aerosol and cloud observations at a mountain site that varies between free tropospheric and boundary layer conditions. The authors employ a new method to separate interstitial aerosol and cloud droplet residuals from a single inlet system in the absence of two separate "total" and CVI inlets by combining effective diameter retrievals and in-cloud size distributions from a DMA. They note the relative uncertainty of the method and quantify penetration fractions of the aerosol. Further, aerosol composition measurements were combined with the size distributions to determine the hygroscopicity and effective supersaturation of interstitial, cloud free, and activated aerosol and these results are compared with a commonly used droplet parameterization where generally good agreement was found between the methods depending on the airmass conditions and activation regime.

Overall, the paper is a very interesting addition to the existing literature and provides an applicable method for distinguishing between droplets and residuals that can be used with previous and future measurements. I only have two major comments and a handful of minor comments, questions, and edits listed below. The paper is generally well written, and the figures provide mostly appropriate information, but I believe a few stray grammatical errors and clarifying captions were not checked prior to submission. I have pointed out a number of these, but I feel several remain, so I ask the authors to please give the paper, figures, and captions a careful re-read prior to resubmission. I believe this paper is appropriate for publication in ACP after these comments and edits are addressed.

**Major Comments:**

If I understand correctly, a Deff minimum threshold of 13.5 µm was applied for all CALISHTO measurements in the "virtual filter" technique and Fig. 3 is only showing this for one case? This raises two questions for me:

1) For all the cloudy measurements considered, did the effective minimum threshold vary case-by-case or was 13.5 µm always the most optimal? Are the authors able to provide a statistical analysis (maybe in the form of a histogram) showing the best minimum threshold (best agreement between integrated concentrations and in situ droplet number) for each case to support why 13.5 µm was selected? This discussion should be included in the main text.

2) 30 min was used as the pre-/in-cloud sampling lengths. Given the scanning time of the SMPS, this would at most be 6 size distributions pre-/in-cloud. Were other sampling lengths tested, why or why not? Do the authors think a different sampling time, and therefore average, would change the results of the optimal minimum threshold?

The authors have done a convincing job arguing that aerosol concentration and type - not changes in cloud dynamics - drive droplet limitation variations. Have the authors considered the impacts of the mean (or modal) sizes of the aerosol populations in the different airmasses and how they may impact hygroscopicity and supersaturation given that size has been shown to be a predominant factor controlling aerosol activation (Dusek et al., 2006; Ovadnevaite et al., 2017)? I would imagine the Hoppel minimum and hygroscopicity near that diameter could impact the activation in different conditions (Dedrick et al., 2024). Given the differences in airmass NTotals, I might also expect mean/modal size to be different for FTL and PBL conditions and impact hygroscopicity (Royalty et al., 2017; Wex et al., 2016). Were FTL aerosol, while less numerous, larger which also made them more hygroscopic? PBL aerosol were more numerous than FTL, where they also smaller which made them less hygroscopic? Illustrating median distributions in FTL and PBL conditions and highlighting these changes, if any, in the shape of the size distributions may provide this insight but may only warrant a brief discussion.

**Minor Comments/Edits:**
Lines 127-128: The authors reference Foskins et al. (under review) in these lines to describe PBLH. Can the authors please provide a brief description of how the PBLH was identified/determined? Was it from radiosondes, lidar, ceilometer, reanalysis, etc...?

Lines 127-135: Are the authors able to provide contextualization of the macrophysical properties of the clouds measured at the site (e.g. cloud type, base height, thickness, etc.) from CALISTHO measurements or previous work (climatology, surface observers)? Were the clouds encountered at the main site typically within or above the PBLH?

Section 2.2 Instrumentation: Have the authors quantified the particle losses to the aerosol measurements? Can these losses be discussed briefly in this section?

Line 238-239: Typo – delete the first "of" in the sentence reading: "...while the hygroscopicity [of] value of eBC was considered..."

Line 294: Typo – "...with [the] observed moisture content..."

Lines 300-301: does Figure 2 show that the winds from the east/southeast are associated with transported dust aerosol, or is this from previous work?

Lines 309-310: Can the authors please check and clarify which figure is being referenced in this discussion? The authors talk about aerosol content (presumably NTotal), Nd, and Deff changes but the figure they are referencing is Fig. S2 which has wind rose plots not Nd or Deff. Are the authors referencing (main text) Figure 2 (please correct the referenced panel letters if so) and/or Figure S3? Further, please be consistent and specific with the terminology of aerosol concentration using NTotal; "aerosol content" is vague.

Figure 2 caption: Because SO42- was also used a proxy for FTL vs. PBL, did the authors see a correlation between SO42- and PBLH? This relationship is a bit difficult to distinguish comparing the time series.

Figure 2: Why are CCN concentration time series not included in this figure? Can the authors please provide this time series either in this plot or as a supplement?

Figure 3: Panel A shows the size distributions averaged for the different Deff thresholds, correct? I don't believe this is clearly specified in the caption or referenced in the text.

Figure 3: In the main text you state that the Deff threshold sensitivity starts at a cutoff of 10 μm (lines 346-347), but the color scale shows cutoffs starting at 0 μm with distributions in panel A and symbols in panel B showing a cutoff Deff <10 μm was used in some cases (blue colors from color bar). Please clarify this discrepancy. Were these lower cut offs shown for comparison/completeness? If so, please state in text and caption.

Figure 3: Please specify in the caption what the solid and dashed/dashed-dotted black lines represent in this figure.

Lines: 324-325: Where in Figure 3 are the authors showing CCN-active aerosol "of order 100 nm" that "grow at least 10-fold"?

Lines 325-328: When the authors state: "when the station was fully covered by clouds…", do they mean the station was "in-cloud" (i.e. LWC>0.02)? In these lines there is a reference to Fig. 2, but that figure does not show "cloudiness" or what is meant by "fully covered by clouds." Fig. S3, however, does separate "cloudiness" and statistics described in this discussion. Did you mean to also reference this figure (Fig. S3)? If Fig. 2 is to be included in this discussion, it would be appropriate for the LWC time series to be added in one of the panels of Fig. 2 (likely panel D). The phrase "fully covered by clouds" is used several times in the manuscript; can this be clarified as "in-cloud" instead?

Lines 329-330: The authors have not shown CCN (timeseries or statistics), therefore the reader cannot glean that the "FTL has fewer CCN" as discussed here. Please provide these metrics.

Line 343: Typo: "We select the periods during we were sampling…" correct to-> "…periods when we were sampling…"?

Line 369: Typo: delete the first "is": "…for this [is] conclusion is…"

Line 374: Typo: check this sentence: "… , include during the…". I think the "include" was left over?

Line 428: Typo: change the first "that" to "what"?

Line 455: Typo: correct "was" to "were" in: "...distributions that [was] measured."

Section 3.4 Closure study of Nd and s*: can the authors please place the retrieved/parameterized Nd and s* in context with previous observations or modelling results? Were these values consistent with previously reported orographic clouds and/or representative of the regional aerosol conditions and airmasses or more similar to other conditions and retrievals?

**References**

Dedrick, J., Russell, L., Sedlacek, A. I., Kuang, C., Zawadowicz, M., & Lubin, D. (2024). Aerosol-Correlated Cloud Activation for Clean Conditions in the Tropical Atlantic Boundary Layer During LASIC. *Geophysical Research Letters, 51*(3). Article.

Dusek, U., Frank, G., Hildebrandt, L., Curtius, J., Schneider, J., Walter, S., et al. (2006). Size matters more than chemistry for cloud-nucleating ability of aerosol particles. *Science, 312*(5778), 1375-1378. Article.

Ovadnevaite, J., Zuend, A., Laaksonen, A., Sanchez, K., Roberts, G., Ceburnis, D., et al. (2017). Surface tension prevails over solute effect in organic-influenced cloud droplet activation. *Nature, 546*(7660), 637-641. Article.

Royalty, T., Phillips, B., Dawson, K., Reed, R., Meskhidze, N., & Petters, M. (2017). Aerosol Properties Observed in the Subtropical North Pacific Boundary Layer. *Journal of Geophysical Research-Atmospheres, 122*(18), 9990-10012. Article.

Wex, H., Dieckmann, K., Roberts, G., Conrath, T., Izaguirre, M., Hartmann, S., et al. (2016). Aerosol arriving on the Caribbean island of Barbados: physical properties and origin. *Atmospheric Chemistry and Physics, 16*(22), 14107-14130. Article.

---

## Author Comment (AC1)

**Responses to Reviewer 1**

Review of "Drivers of Droplet Formation in East Mediterranean Orographic Clouds" by Foskinis et al. Summary: "This paper investigates … prior to resubmission. I believe this paper is appropriate for publication in ACP after these comments and edits are addressed.

**Reply:** We thank the reviewer for their enthusiam of this work, as well as for all the comments that have lead to an improved manuscript. We have addressed each point raised and have also reread the manuscript to correct for any stray grammatical errors.

**Major Comments**:

If I understand correctly, a Deff minimum threshold of 13.5 µm was applied for all CALISHTO measurements in the "virtual filter" technique and Fig. 3 is only showing this for one case?

**Reply:** Figure 3 presents the sensitivity analysis, based on **all** the cloud cases that which we were able to capture the cloud transition where the site was sampled for at least 30 minutes continuously in cloud-free conditions followed by (or proceeded by) at least 30 minutes of cloudy conditions, while each curve has derived from the averaged of all those cases using differential cut-off size thresholds. Thus, it is not only one of the cases. We apologize for this misunderstanding and will make this very clear in the caption and the text.

o   This raises two questions for me:
o   1) For all the cloudy measurements considered, did the effective minimum threshold vary case-by-case or was 13.5 µm always the most optimal? Are the authors able to provide a statistical analysis (maybe in the form of a histogram) showing the best minimum threshold (best agreement between integrated concentrations and in situ droplet number) for each case to support why 13.5 µm was selected? This discussion should be included in the main text.

**Reply:** Given that the sensitivity analysis was applied over all the cloud event cases, it represents a global threshold, and its validity was confirmed with the analysis of Figure 3b – as the 13.5 µm threshold provides agreement (on average) between the observed droplet number and the integrated difference of the pre-cloud and in-cloud aerosol (which corresponds to the activated droplets). The errorbars represent the distribution of the uncertainty around these data points. Figure 3b also shows the closure variability/uncertainty (expressed by the error bars) grows significantly as you move away from the 13.5 µm optimal threshold. We will include these clarifications in the main text.

o   2a) 30 min was used as the pre-/in-cloud sampling lengths. Given the scanning time of the MPSS, this would at most be 6 size distributions pre-/in-cloud. Were other sampling lengths tested, why or why not?

**Reply:** This is a good point! We tested a number of different sampling times (not shown), and what we found was that by decreasing sampling time the closure becomes noisier (which is expected) while by increasing it, the number of cases for closure decreases significantly becuase clouds with a lifetime smaller than the sampling time are excluded (the requirement is that a cloud is continuously over the side during each sampling period). Thus, the 30-minute sampling time was found a "best choice" to obtain a reasonable number of samples that at the

same time are subject to the least statistcal noise from the natural variability occuring in each cloud event. We will include these clarifications in the main text.

○ 2b) Do the authors think a different sampling time, and therefore average, would change the results of the optimal minimum threshold?

**Reply:** This is a good point as well. Looking at Figure 3b the optimum cut-off is subject to some uncertainty which could change by the sampling time. We did not explore this however because 30 minutes is the chosen sampling time for the reasons mentioned before. This could be the topic of future research

○ 3) The authors have done a convincing job arguing that aerosol concentration and type - not changes in cloud dynamics - drive droplet limitation variations. Have the authors considered the impacts of the mean (or modal) sizes of the aerosol populations in the different airmasses and how they may impact hygroscopicity and supersaturation given that size has been shown to be a predominant factor controlling aerosol activation (Dusek et al., 2006; Ovadnevaite et al., 2017)? I would imagine the Hoppel minimum and hygroscopicity near that diameter could impact the activation in different conditions (Dedrick et al., 2024). Given the differences in airmass NTotals, I might also expect mean/modal size to be different for FTL and PBL conditions and impact hygroscopicity (Royalty et al., 2017; Wex et al., 2016). Were FTL aerosol, while less numerous, larger which also made them more hygroscopic? PBL aerosol were more numerous than FTL, where they also smaller which made them less hygroscopic? Illustrating median distributions in FTL and PBL conditions and highlighting these changes, if any, in the shape of the size distributions may provide this insight but may only warrant a brief discussion.

**Reply:** We do not carry out modal fits, but use the original MPSS (bin) data together with size-resolved hygroscopicity when (HAC)[2] was sampling within Cloud-Free and in-Cloud regimes (the latter was examined separately using the virtual filtering technique to separate the interstitial from the mixture regimes). What we found is that smaller particles tend to be more hygroscopic than the larger ones during cloud events – and this contrast increases when sampling a mixture of interstitial and dried droplets. Given that the interstial aerosol is less hygroscopic (Figure S6b), the high value of hygroscopicity shown in Figure S6c, especially when the airmass originates from the PBL, are related to evaporated cloud droplets. Additionally, Figure S6d reveals that the activated aerosols, when originating from the FT, are smaller in size compared to particles from the BL. Thus, the modal/mean sizes of the aerosol populations are linked to changes in hygroscopicity. The modified text now summarises these points.

**Minor Comments/Edits:**

○ Lines 127-128: The authors reference Foskinis et al. (under review) in these lines to describe PBLH. Can the authors please provide a brief description of how the PBLH was identified/determined? Was it from radiosondes, lidar, ceilometer, reanalysis, etc…?

**Reply:** Done. Now it is updated including the following description "…, here we used the PBLH data that were derived according to Foskinis et al. (under review) based on the turbulence threshold technique which was applied on the wind Doppler lidar measurements."

○ Lines 127-135: Are the authors able to provide contextualization of the macrophysical properties of the clouds measured at the site (e.g. cloud type, base height, thickness, etc.) from CALISTHO measurements or previous work (climatology, surface observers)? Were the clouds encountered at the main site typically within or above the PBLH?

**Reply:** We usually sample orographic warm clouds with a cloud base that can vary with respect to the (HAC)2 level, which depending on its relation to the PBLH can form on PBL or FTL airmasses. The clouds are usually a few hundred meters thick, as evidenced by the LWC and the droplet size measurements. We can also have multi-layer clouds depending on the meteorological regimes but those are usually not the cases that we use here. A full analysis of cloud statistics based on the radar and other remote sensing instruments is the topic of a future study.

o Section 2.2 Instrumentation: Have the authors quantified the particle losses to the aerosol measurements? Can these losses be discussed briefly in this section?

**Reply:** We apologize for that, now we have updated Section 2.2.2 including the details about the corrections for the particle losses.

o Line 238-239: Typo – delete the first "of" in the sentence reading: "…while the hygroscopicity [of] value of eBC was considered…"
Done

o Line 294: Typo – "…with [the] observed moisture content…"
Done

o Lines 300-301: does Figure 2 show that the winds from the east/southeast are associated with transported dust aerosol, or is this from previous work?
**Reply:** This is shown by the companion study by Gao et al. (2024), and also is supported by Stavroulias et al (2021), as common synoptic patterns favour the transport of dust aerosols.

o Lines 309-310: Can the authors please check and … is vague.
**Reply:** We apologize for this lack of clarity, there was a mistake in labelling which was also mentioned by Reviewer 2, now we have corrected the manuscript properly including the same terminology as mentioned above.

o Figure 2 caption: Because SO42- was also used as a proxy for FTL vs. PBL, did the authors see a correlation between SO42- and PBLH? This relationship is a bit difficult to distinguish comparing the time series.
**Reply:** The SO42 was used only as a qualitative proxy because it went below the detection limits of the ACSM when the (HAC)2 was within the FTL. Thus, something more quantitative is unfortunately not feasible.

o Figure 2: Why are CCN concentration time series not included in this figure? Can the authors please provide this time series either in this plot or as a supplement?
**Reply:** Good point! We now provide the CCN timeseries at three supersaturation values (s=0.1,0.3,0.7%) in Figure 2f and the LWC in Figure 2e as mentioned below.

o Figure 3: Panel A shows the size distributions averaged for the different Deff thresholds, correct? I don't believe this is clearly specified in the caption or referenced in the text.

**Reply:** Now it is reads as "a) The averaged $dN/dlogD_p$ size distribution of the captured cloud transition moments when using different cut-off threshold value so-called "sensitivity analysis" and the cut-off threshold ($D_{eff}$ as measured *in-situ* with the PVM-100) that was applied is represented by the color-scale. b) The integrated difference between pre-cloud and in-cloud aerosol size distributions from ~70 nm to the largest sizes when compared to the droplet number measured *in-situ* concurrently by the PVM-100. Each symbol corresponds to the application of a different $D_{eff}$ threshold (as indicated by the symbol color, using the same scheme as in a), while the errorbars correspond to the standard deviation."

o   Figure 3: In the main text you state that the Deff threshold sensitivity starts at a cutoff of 10 μm (lines 346-347), but the color scale shows cutoffs starting at 0 μm with distributions in panel A and symbols in panel B showing a cutoff Deff 0.02)? In these lines there is a reference to Fig. 2, but that figure does not show "cloudiness" or what is meant by "fully covered by clouds."
    **Reply:** We thank the reviewer for pointing out this mistake, which is now corrected. The second comment refers to Lines 335-336, " However, $D_{eff}$ varies considerably during a cloud event (**Error! Reference source not found.**), and often exceeds 10 μm.", while now changed to " However, $D_{eff}$ varies considerably during a cloud event (**Error! Reference source not found.**e), and often exceeds 10 μm.", where the measurement of $D_{eff}$ refers to in-cloud moments.

o   Fig. S3, however, does separate "cloudiness" and statistics described in this discussion. Did you mean to also reference this figure (Fig. S3)? If Fig. 2 is to be included in this discussion, it would be appropriate for the LWC time series to be added to one of the panels of Fig. 2 (likely panel D).
    **Reply:** What is mentioned here is referred to in Lines 309-310, now it is updated as "We observed that the increase of $N_{Total}$ (from ~250 cm$^{-3}$ to ~750 cm$^{-3}$) (Figure S3a) leads to an increase of $N_d$ (from ~100 cm$^{-3}$ to ~300 cm$^{-3}$) (Figure S3b), and decrease of the cloud droplet size ($D_{eff}$) (from ~17.5 μm to ~10 μm) (Figure S3c), consistent with the Twomey effect (Twomey, 1977) of aerosols on clouds and cloud albedo (IPCC, 2023)." Additionally, the LWC timeseries now it is included in Figure 2e.

o   The phrase "fully covered by clouds" is used several times in the manuscript; can this be clarified as "in-cloud" instead?
    Done

o   Lines 329-330: The authors have not shown CCN (timeseries or statistics), therefore the reader cannot glean that the "FTL has fewer CCN" as discussed here. Please provide these metrics.
    **Reply:** The CCN timeseries now it is included in Figure 2f.

o   Line 343: Typo: "We select the periods during we were sampling…" correct to-> "…periods when we were sampling…"?
    Done

- Line 369: Typo: delete the first "is": "…for this [is] conclusion is…"
  Done

- Line 374: Typo: check this sentence: "… , include during the…". I think the "include" was left over?
  Done

- Line 428: Typo: change the first "that" to "what"?
  Done

- Line 455: Typo: correct "was" to "were" in: "…distributions that [was] measured."
  Done

- Section 3.4 Closure study of Nd and s*: can the authors please place the retrieved/parameterized Nd and s* in context with previous observations or modelling results? Were these values consistent with previously reported orographic clouds and/or representative of the regional aerosol conditions and airmasses or more similar to other conditions and retrievals?
  This information and comparison are included in the overview paper.

---

## Author Comment (AC2)

**Responses to Reviewer 2**

Review of "Drivers of Droplet Formation in East Mediterranean Orographic Clouds" by Foskinis et al. Summary: *"The manuscript deals with ...* The manuscript is generally well written; however, Section 3.4 deserves more focus as it contains some obscure arguments.

**Reply:** We thank the reviewer for their enthusiam of this work, as well as for all the comments that have lead to an improved manuscript. We have addressed each point raised and have also reread the manuscript to correct for any stray grammatical errors.

**Specific comments:**

o Section 2.3.1: When determining the hygroscopicity kappa value starting from the chemical composition, what factor has been applied to the organic mass fraction?

**Reply:** The hygroscopicity value of *eBC* was considered equal to 0.2 based on Ding et al. (2021)

o "we identified three prevailing wind directions, that correspond to the local transport patterns (Figure S2f) from 90°, 180° and 320° N, where the NTotal obtains its maximum values (~3300 cm-3)". It is not clear whether the maximum values for NTotal are found only at the 320° direction or all the three selected wind directions.

**Reply:** We apologize for the lack of clarity. We have now updated Figure S2f including the boxplots of the $N_{Total}$ related to the wind direction and discussed in the manuscript.

Line 306: It is not really clear in Figure S2 what are the wind directions tracing an air flow passing over mountain peaks. In addition, with wind directions from 330 – 360°, the PBL height is large and vertical velocity as well (Fig S2b) which is the other way around with respect to what stated in the text.

**Reply:** Actually, here the reviewer is mentioning the same thing with what is shown in the manuscript. When airflow originates from 330 – 360°, in principle, the airmass passes over mountain peaks, increasing PBLH values. In contrast, when the wind blows from SW or NE directions, where there are no mountain peaks among the trail, the PBL tends to shrink. Thus, both of us claim the same thing, so no change is needed.

o Lines 308 – 310: the references to the panels in Fig. S2 (panels b, c, d) in the text do not match with what shown in Fig. S2 in the supplementary. Are the Authors referring to Fig S3 instead?

**Reply:** This point has been mentioned by Reviewer 1, now it is corrected properly.

o Fig. 2. Panel b: the PBL / FTL mask is not clear; for instance, on days 11-14 Nov the PBLH is lower than the (HAC)2 elevation but this period is classified green (PBL); the same occurs on 20 – 23 Nov.

**Reply:** The reviewer is right, and we apologize for this oersight. The graph is now correct.

o Panel c: In the wind direction plot, please use a palette with colours changing with continuity between 360° and 0°.

Done

o    Fig. 3a: the y axis of the figure on the left cannot report simply concentrations in cm-3 but must be in dN/dlogDp units. The specific period of the campaign providing this subset of data should be reported. Same for Fig. 5.

**Reply:** The figure corresponds to the whole period of the campaign as mentioned above, while we changed Figures 3 and 5 as requested.

o    About the "virtual cutoff". Clearly, a (physical) PM10 cutoff … exceeding the threshold of 13.5 μm" (lines 431 - 433). However, with Deff < 13.5 μm, the interstitial dataset should be contaminated from the concentrations of droplet residuals. Please, explain.

**Reply:** We apologize for this misunderstanding, there was a typo in the text. We meant to write that " where the $D_{eff}$ was **not** exceeding the threshold of 13.5 μm". Everything is now corrected.

o    Section 3.4: if clouds are observed only at the (HAC)2 site and not at VL, why the values in the y axes differ between the two graphs in Fig. 6? Most noticeably, in the FTL conditions, Deff is larger than 13.5 μm and the PM10 inlet at (HAC)2 site should sample interstitial aerosols free of cloud residuals, so why the closure is worse in this case? Please, clarify.

**Reply:** In the case of FTL, where as correctly mentioned the sampled aerosols correspond only to the interstitials, there is a loss of aerosols (those which have activated into cloud droplets), that are not accounted for by the parameterization calcualtions. Thus, the predicted droplet number is much smaller than the observed.